# Amplification of global warming through pH-dependence of DMS-production simulated with a fully coupled Earth system model

Jörg Schwinger[1], Jerry Tjiputra[1], Nadine Goris[1], Katharina D. Six[2], Alf Kirkevåg[3], Øyvind Seland[3], Christoph Heinze[4,1], and Tatiana Ilyina[2]

[1]Uni Research Climate, Bjerknes Centre for Climate Research, Bergen, Norway
[2]Max Planck Institute for Meteorology, Hamburg, Germany
[3]Norwegian Meteorological Institute, Oslo, Norway
[4]Geophysical Institute, University of Bergen, Bjerknes Centre for Climate Research, Bergen, Norway

*Correspondence to:* J. Schwinger (jorg.schwinger@uni.no)

**Abstract.** We estimate the additional transient surface warming $\Delta T_s$ caused by a potential reduction of marine dimethyl sulfide (DMS) production due to ocean-acidification under the high emission scenario RCP8.5 until the year 2200. Since we use a fully coupled Earth system model, our results include a range of feedbacks, such as the response of marine DMS-production to the additional changes in temperature and sea-ice cover. Our results are broadly consistent with the findings of a previous study that employed an off-line model set-up. Assuming a medium (strong) sensitivity of DMS-production to pH, we find an additional transient global warming of 0.30 K (0.47 K) towards the end of the 22nd century when DMS-emission are reduced by 7.3 Tg S yr$^{-1}$ or 31% (11.5 Tg S yr$^{-1}$ or 48%). The main mechanism behind the additional warming is a reduction of cloud albedo, but a change in short-wave radiative fluxes under clear-sky conditions due to reduced sulfate aerosol load also contributes significantly. We find an approximately linear relationship between reduction of DMS-emissions and changes in top of the atmosphere radiative fluxes as well as changes in surface temperature for the range of DMS-emissions considered here. For example, global average $T_s$ changes by $-0.041$ K per 1 Tg S yr$^{-1}$ change in sea–air DMS-fluxes. The additional warming in our model has a pronounced asymmetry between northern and southern high latitudes. It is largest over the Antarctic continent, where the additional temperature increase of 0.56 K (0.89 K) is almost twice the global average. We find that feedbacks are small on the global scale due to opposing regional contributions. The most pronounced feedback is found for the Southern Ocean, where we estimate that the additional climate change enhances sea–air DMS-fluxes by about 9% (15%), which counteracts the reduction due to ocean acidification.

## 1 Introduction

Changes in emissions of marine dimethyl sulfide (DMS) have the potential to influence climate via a modification of aerosol and cloud properties. The implications of a DMS climate feedback were first described by Shaw (1983) and Charlson et al. (1987). The latter authors hypothesise that DMS-production and emission, the number of cloud-condensation nuclei (CCN), and the albedo of marine boundary layer clouds are interlinked in a negative feedback loop acting to stabilize the Earth's

climate against external perturbations. This idea has become known as the CLAW-hypothesis (after the initials of the authors, Charlson, Lovelock, Andreae, and Warren).

DMS is a by-product of marine primary production (e.g. Liss et al., 1994) and it is the main natural source of atmospheric sulfur (about $28\,\text{Tg}\,\text{S}\,\text{yr}^{-1}$, Lana et al., 2011). Once in the atmosphere, DMS is oxidised to $SO_2$ and methanesulfonic acid (MSA), and further to gaseous $H_2SO_4$, which rapidly condenses onto pre-existing aerosol particles or, in the absence of sufficient aerosol surface area, forms nucleation mode sulfate particles (Carslaw et al., 2010). Hence, sulfate produced from DMS leads to both increased numbers of nucleation mode sulfate and increased size and modified hygroscopicity of larger sulfate and other internally mixed particles. Korhonen et al. (2008) show that, although nucleation mode sulfate particles do not directly contribute to CCN, the formation and subsequent growth of these particles in the free troposphere can be the main source of DMS-derived CCN in remote ocean areas. Sulfate from condensation can increase CCN numbers in areas where the pre-existing particles are too small or too hydrophobic, but may in other areas or situations contribute to a reduction of the CCN production, if pre-existing particles are already large enough to activate or more hygroscopic than sulfate.

The effect of anthropogenic climate change on DMS-production and emission has been shown to be relatively small globally (Bopp et al., 2003; Gabric et al., 2004; Gunson et al., 2006; Vallina et al., 2007; Kloster et al., 2007), although, regionally, larger sensitivities have been reported (Bopp et al., 2003; Kloster et al., 2007). Most of these studies predict an increase of global surface DMS-concentrations or sea–air fluxes of about 2 to 14% in response to global warming, while Kloster et al. (2007) model a 10% decrease under a future scenario. It has been further shown that the sensitivity of CCN numbers to changes in DMS-emissions is relatively low (Woodhouse et al., 2010) such that there is currently no evidence for a significant CLAW-like feedback on the global scale.

More recently, results from mesocosm studies have indicated that DMS-production in marine ecosystems might decrease with decreasing seawater pH (e.g., Archer et al., 2013; see also the discussion in the supplementary material of Six et al., 2013, and references therein). These findings were confirmed by follow up mesocosm studies in warmer ocean regions such as the coastal waters of Korea or in the sub-tropics off the Canaries (Park et al., 2014, Archer, pers. comm.).

Six et al. (2013) study the possible impacts of this phenomenon on the global radiation balance. Assuming a low, medium, and high sensitivity of DMS-production to pH, they model the decrease of DMS sea–air fluxes due to warming and progressing ocean acidification for the A1B scenario from the Special Report on Emissions Scenarios (SRES). They find a global reduction of DMS-emissions due to ocean acidification (in addition to reductions due to climate change) of 11% in the medium-sensitivity run (19% for the high-sensitivity). In a separate step, they feed the reduced marine DMS-emissions into an atmospheric general circulation model with aerosol-chemistry to derive an effective radiative forcing, and, finally, to estimate a range of additional equilibrium warming that results from this forcing (0.23–0.48 K for the medium pH-sensitivity). Due to the off-line design of the Six et al. (2013) study, some important aspects of the DMS sea–air flux reduction are not accessible, for example, the spatial pattern of the additional warming, as well as any feedback between atmospheric changes and DMS-production itself.

The purpose of this study is to simulate the possible link between ocean-acidification, sea–air DMS-fluxes, and changes in surface climate in a fully coupled Earth system model that includes a prognostic marine DMS-scheme as well as an aerosol module capable of modelling changes of the radiation balance due to altered atmospheric sulfur cycling. The marine biogeo-

chemistry model employed in this study is very similar to the one used by Six et al. (2013), with only minor differences (see Sect. 2.1). The prognostic DMS scheme as well as the assumptions on the pH-dependency of DMS-production are the same as in Six et al. (2013). We simulate the transient climate change and its amplification by pH-dependence of DMS-production under the RCP8.5 scenario and its extension to the year 2200 (Meinshausen et al., 2011). This scenario and the simulation period up to 2200 is chosen to get a clear signal of transient change. For the same reason, we focus our analysis on the medium and strong sensitivity of DMS-production to pH, without implying that the observational evidence is better for these than for a lower sensitivity.

We first present a brief description of our Earth system model and of the experimental set-up in Sect. 2. An evaluation of sea–air DMS-fluxes and its drivers as well as the changes found in the baseline RCP8.5 simulation (that assumes no pH-dependency of DMS-production) is given in Sect. 3.1. We then present the results of two pH-sensitive simulations (medium and high sensitivity) and an analysis of feedbacks in Sect. 3.2 and 3.3, respectively. A summary of our results and our conclusions are provided in Sect. 4.

## 2   Model description and experimental set-up

The Norwegian Earth System model NorESM1-ME employed in this study is based on the Community Earth System Model (CESM1-BGC, Gent et al., 2011; Lindsay et al., 2014). It uses the same sea-ice and land models, but a different ocean component and a different aerosol module embedded in the atmosphere model. Here, we use the model version that participated in the Coupled Model Intercomparison Project Phase 5 (CMIP5, Taylor et al., 2012), and that has been described and evaluated in a series of papers (Bentsen et al., 2013; Iversen et al., 2013; Tjiputra et al., 2013; Kirkevåg et al., 2013). The only difference between the published model version and the model version employed for this study, is the coupling of prognostic sea–air DMS-fluxes calculated by the ocean biogeochemistry component to the atmosphere model (as opposed to the use of prescribed climatological DMS-emissions in the CMIP5 model version).

We do not give an extensive model description here, but refer the reader to the publications cited above. In the following sections, we provide a brief description of the marine biogeochemistry and aerosol modules with focus on the parameterisations relevant for this study.

### 2.1   The marine biogeochemistry model MICOM-HAMOCC

The physical ocean component of NorESM is the isopycnic MICOM (Miami Isopycnic Coordinate Ocean Model, Bleck et al., 1992), albeit with considerable modifications of numerics and physics as described in Bentsen et al. (2013). The HAMburg Ocean Carbon Cycle model (HAMOCC, Maier-Reimer, 1993; Maier-Reimer et al., 2005) has been implemented into MICOM by Assmann et al. (2010). The HAMOCC version used in NorESM1-ME is further described by Tjiputra et al. (2013) and Schwinger et al. (2016). We note that HAMOCC, since it was implemented into MICOM, has also been further developed by the biogeochemistry group at the Max Planck Institute in Hamburg. As a result, our model version is very similar but not identical to the one described by Ilyina et al. (2013). For example, the inorganic carbon chemistry has been updated for our

model version and some technical modifications were necessary for the implementation into the isopycnic MICOM. Further, some parameters were differently tuned because of the different physical ocean models (see Schwinger et al., 2016, for details). None of these differences is expected to alter the findings and conclusions of this study.

HAMOCC simulates the marine biogeochemical cycles of carbon, phosphorous, nitrogen, silica, and iron. Biological production and export out of the euphotic zone is parameterised by a NPZD-type (Nutrient-Phytoplankton-Zooplankton-Detritus) ecosystem model extended to include dissolved organic carbon (Six and Maier-Reimer, 1996). The model has only one generic phytoplankton and one generic zooplankton type. Bacteria are not modeled explicitly, rather, processes related to bacterial activity (e.g. remineralisation of dissolved and particulate organic matter) are assumed to proceed at constant rates. A constant Redfield ratio of 1:16:122 (P:N:C) is used for the composition of organic matter. Primary production (PP) is limited by the least available macronutrient (phosphate, nitrate) or micronutrient (iron). Detritus is formed through grazing activity as well as phytoplankton and zooplankton mortality. Once formed, it sinks through the water column at a constant speed of $5\,\mathrm{m\,d^{-1}}$ and is remineralised at a constant rate. Although calcifying and silicifying plankton functional types are not modeled explicitly, the fraction of calcium carbonate and biogenic silica that is added to the pool of sinking shell material depends on the availability of silicic acid. Here, it is implicitly assumed that diatoms out-compete other phytoplankton species when the supply of silicic acid is ample. This is parameterised by assuming that a fraction $[\mathrm{Si}]/(K_{\mathrm{Si}}+[\mathrm{Si}])$ of detritus production contains opal shells while the remaining fraction contains calcareous shells, where $K_{\mathrm{Si}} = 1\,\mathrm{mmol\,Si\,m^{-3}}$ is the half-saturation constant for silicate uptake. We note that HAMOCC's ecosystem model has no dependency on inorganic carbon availability or pH, e. g., neither cell carbon quotas or stoichiometric ratios nor calcification rates are assumed to vary with changing pH.

The parameterisation of the DMS-cycle in HAMOCC has been implemented and evaluated by Six and Maier-Reimer (2006) and Kloster et al. (2006). Processes included are DMS-production and losses by bacterial consumption, photolysis, and sea–air DMS gas exchange. Observations indicate that when the cell membrane of phytoplankton is disrupted by senescence or due to viral attack and zooplankton grazing, the DMS-precursor dimethylsulphoniopropionate (DMSP) is released into sea water and rapidly converted to DMS by bacterial and algal enzymes (Stefels et al., 2007). Therefore, the DMS-production in our model is assumed to be a function of detritus production. It is further modified by the production of opal and $\mathrm{CaCO_3}$ shell material, that is, calcite or opal producing organisms are assumed to have different sulfur to carbon ratios. This differentiation credits the fact that haptophytes have, in general, a higher DMSP to cell carbon ratio and, thus, have a higher contribution to DMS-production (Keller et al., 1989). Bacterial consumption is a linear function of temperature and a monotypic saturation function of DMS-concentration. Photolysis and sea–air gas exchange are linear functions of the seawater DMS-concentration. The local photolysis rate depends on the intensity of incoming light, and the calculation of sea–air DMS-fluxes follows the gas-exchange parameterisation of Wanninkhof (1992). For this study, we use the same set of tunable parameters as Kloster et al. (2006) and Six et al. (2013), except for the scaling factor for bacterial consumption, which is reduced by half to better reproduce the observed fluxes. We refer the reader to Kloster et al. (2006) and Six et al. (2013) for a detailed description of the DMS-parameterisation used here.

The applied relationship between DMS-production and seawater pH is motivated by a compilation of the results of several mesocosm studies (Six et al., 2013). In these mesocosm experiments, the temporal evolution of DMS-concentration within a

confined natural water volume is investigated under different levels of $CO_2$ partial pressure with corresponding seawater pH (further information on the method is given in Archer et al., 2013). As in Six et al. (2013) we assume a linear decrease of DMS-production ($P_{DMS}$) with decreasing pH (Fig. 1 in Six et al., 2013), by multiplying $P_{DMS}$ with a factor $F = 1 + (pH_{pi} - pH)\gamma$. Here, pH is the modeled local pH-value, and $pH_{pi}$ is the local pH of the pre-industrial undisturbed ocean, relative to which a deviation of pH is measured. $pH_{pi}$ is taken from a monthly climatology calculated from 10 years of a preindustrial control simulation. The constant $\gamma$ defines the sensitivity of DMS-production to pH, and is chosen following Six et al. (2013, see Sect. 2.3). We note that the reason for the observed decrease in DMS-concentrations under low pH is still debated. Shifts in species composition seem to play a major role in general (Archer et al., 2013; Park et al., 2014), but also changes in the rate of DMSP to DMS conversion and the rate of loss through bacterial consumption might contribute to the observed pH-dependency. All these processes cannot be resolved explicitly by our model, and, as a first approach, we assign a linear pH-dependency to DMS-production.

## 2.2 The aerosol module of CAM4-Oslo

As described in detail by Kirkevåg et al. (2013) and Seland et al. (2008), the aerosol life cycle scheme in the atmosphere component of NorESM1-ME (Community Atmosphere Model Version 4-Oslo, CAM4-Oslo) calculates and traces, for each constituent, aerosol mass mixing ratios which are tagged according to production mechanisms in clear and cloudy air. The processes treated in the model are gas phase and aqueous phase chemical production, gas to particle production (nucleation), condensation, and coagulation of small particles onto larger pre-existing particles. Primary particles are emitted as accumulation mode sulfate, nucleation and accumulation mode black carbon (BC), Aitken mode BC, internally mixed Aitken mode organic matter (OM) and BC, Aitken, accumulation, and coarse mode sea salt, and accumulation and coarse mode mineral dust.

Aerosol precursor gas-phase components accounted for are DMS and $SO_2$. DMS is produced in the surface ocean and emitted to the atmosphere as described above. It is depleted by oxidation to particulate methane sulfonic acid (MSA), thenceforth treated as primary ocean-biogenic OM, and by oxidation to $SO_2$. Oxidant fields (OH, $O_3$, and $H_2O_2$) for the sulfur chemistry are prescribed as detailed in Kirkevåg et al. (2013). Apart from being produced from DMS, $SO_2$ has important natural sources from volcanoes, and is otherwise emitted from combustion of fossil fuel and biomass containing sulfur. Gaseous sulfate (as $H_2SO_4$) produced in air by oxidation of $SO_2$ by OH is allowed to condense on pre-existing particles whenever sufficient particle surface area is available for condensation. Whatever gaseous sulfate is left after condensation (during a model time step) is assumed to form nucleation mode sulfate by gas to particle production.

Internally mixed water from condensation of water vapour is treated separately through use of look-up tables (calculated off-line) for optical parameters, accounting for the above listed processes as well as hygroscopic swelling. Another set of look-up tables is used to obtain dry size parameters (dry radius and standard deviation) of the aerosol population, which are used as input in the calculation of CCN activation following Abdul-Razzak and Ghan (2000). Aerosol components dissolved in cloud water are not kept as separate tracked variables but are either scavenged or distributed to accumulation mode sulfate as well as accumulation and coarse mode particles in internal mixtures.

## 2.3 Experimental set-up

We have run three sets of simulations (in addition to a control simulation with constant pre-industrial settings, Table 1) that assume different sensitivities of DMS-production to ocean acidification. Each set consists of a historical simulation (1850 to 2005) followed by a RCP8.5 scenario simulation (2006 to 2100) and its extension to the year 2200. The first set of runs does not assume any sensitivity of DMS-production to pH ($\gamma = 0$) and we refer to these simulations collectively as the "experiment BASE". Following Six et al. (2013), the second and third set of model runs assume a medium (experiment SMED, $\gamma = 0.58$) and a high (experiment SHIGH, $\gamma = 0.87$) sensitivity of DMS-production to pH. We use the emission-driven configuration of NorESM, that is, atmospheric $CO_2$ concentrations evolve freely in response to anthropogenic emissions as well as ocean and land carbon sinks (the latter two being potentially different for each experiment).

Prior to these experiments, the NorESM1-ME was spun up for 900 years with a prescribed atmospheric $CO_2$ concentration fixed at 284.7 ppm. Subsequently, the model was further spun up for 550 years in its emission driven configuration with prescribed zero emissions. Towards the end of the second spin-up period, the air–sea and air–land $CO_2$ fluxes closely balance one another and atmospheric $CO_2$ is stable at 284 ppm.

The control simulation is used to account for remaining model drift. Most importantly, there is a small decrease in primary production of about $0.57 \, \mathrm{Pg \, C \, century^{-1}}$, and a corresponding decrease in sea–air DMS-fluxes of $0.51 \, \mathrm{Tg \, S \, century^{-1}}$, due to the fact that fluxes of nutrients to the sediment are not replenished by any mechanism in the model version employed for this study. Note that there is almost no drift ($< 0.01 \, \mathrm{Pg \, C \, century^{-1}}$) in $CO_2$-fluxes due to compensating effects in carbon and $CaCO_3$ export production. Since we wish to conserve the internal consistency of model fields, we do not correct for model drift. Rather, we express all results either relative to the control simulation or relative to the experiment BASE.

## 3 Results and discussion

To begin with, we examine changes of primary production and DMS sea–air fluxes in the BASE simulation relative to the pre-industrial control run. Since, in our model, DMS-production is tied to detritus production, which very closely follows PP, patterns of changes in PP are an useful indicator of changes in DMS-production (with some modifications discussed below). In experiment BASE, these changes are caused by climate change alone (no influence of ocean acidification, since $\gamma = 0$). Additional changes relative to the BASE experiment that are caused by the pH-sensitivity of DMS-production in SMED and SHIGH will be analyzed and discussed in Sects. 3.2 and 3.3.

### 3.1 Primary production and DMS-emissions in experiment BASE

Results for the simulation BASE are summarised in Table 2. During the last 30 years of the 21st and 22nd century, atmospheric $CO_2$ concentrations reach 945 and 1951 ppm, respectively. These numbers are higher than the corresponding prescribed concentrations of RCP8.5, since our model shows a rather low carbon uptake by the land biosphere (Arora et al., 2013) due to the

inclusion of nitrogen limitation of plant productivity. The global average surface air temperature increase ($\Delta T_s$) simulated by our model is 4.2 and 8.1 K for the time periods 2071 to 2100 and 2171 to 2200, respectively.

The spatial pattern of annual primary production in the control run and an estimate of PP derived from MODIS observations and three different processing algorithms (see Schwinger et al., 2016, for details) are shown in Fig. 1a and b. The spatial pattern of PP is well reproduced by the model, although PP is higher than the observation-based estimate in the equatorial Pacific upwelling region as well as south of 40°S. Lower than observed values are found in the subtropical gyres of the Pacific and the whole low-latitude western Pacific as well as in large parts of the Indian Ocean. The resulting annual globally integrated PP (44.2 Pg C in the control run) is lower than the satellite based PP estimate (59.9 Pg C).

Under the RCP8.5 scenario, PP declines globally by 4.2 Pg C towards the end of the 21st century (Table 2, Fig. 1c) and by 5.3 Pg C towards the end of the 22nd century (Table 2, Fig. 1d). This decline is due to increased stratification (reduced mixed layer depth, not shown) and less nutrient supply almost everywhere north of 40°S and south of the seasonally ice covered parts of the North Atlantic and North Pacific. One notable exception is the south-eastern part of the Pacific, where increased mixing by surface winds outweighs the increased thermal stratification in our model, leading to increased nutrient supply and PP. Arctic regions that are seasonally or permanently ice covered in the control run, experience an increase in PP due to reduction in both light and temperature limitations. In the Southern Ocean south of 40°S, we can identify three more or less distinct zones. In the southernmost part, along the coast of Antarctica, declining sea ice leads to an increase in PP. North of the control run sea ice edge, we find a belt where PP partially decreases, while PP increases quite substantially in the northernmost part of the Southern Ocean. This pattern is caused by a southward shift of the circumpolar storm track (and hence cloud cover), which causes increased light limitation in the southern part of the Southern Ocean in our simulations. In the northern part of the Southern Ocean around 50°S increasing temperature and less light limitation sustain a substantially stronger PP under climate change.

We note that the pattern of PP changes found in our model is consistent with the results from Six et al. (2013) and with results from other state-of-the-art ESMs (Steinacher et al., 2010; Bopp et al., 2013; Laufkötter et al., 2015). Although this is also true for the Southern Ocean, the increase of PP south of 40°S is, compared to other models, large relative to decreases elsewhere. This might indicate that the higher than observed PP south of 40°S also implies a relatively high sensitivity to climate change.

Modeled DMS surface concentrations for boreal winter (DJF) and summer (JJA) and the corresponding climatologies from Lana et al. (2011) are shown in Fig. 2. The too high and too low productivity in the eastern equatorial Pacific and the Indian Ocean, respectively, are visible in the surface DMS concentrations in these regions year round. The elevated DMS production around Antarctica in austral summer is well reproduced by our model, except for regions with too extensive summer sea ice cover (Wedell Sea and the eastern part of the Ross Sea). The zonal mean DJF DMS-concentration south of 60°S is therefore significantly higher in the observation-based climatology than in our model (Fig. 2e). Due to too low PP in our model, DMS concentrations approach zero during winter poleward of 40°N and 40°S, while observations indicate winter values around 1 $\mu$ mol m$^{-3}$ in the zonal mean.

The change of DMS-fluxes to the atmosphere in the RCP8.5 scenario simulation (Fig. 3) mainly follows the change in primary production. This is to be expected, since, in our model, detritus production (to which DMS-production is tied) very

closely follows PP on an annual time-scale (about 20% with little spatial variability). However, since the DMS-production is modified by the fraction of opal and calcium carbonate contained in detritus, there are a few subtle differences. First, around Antarctica, the increase in DMS-emissions towards the end of the 22nd century due to loss of sea ice is less than one would infer from PP increase alone. In this region, there is ample supply of silica and the increase in production is caused mainly by an increase of opal producing organisms. Second, in latitudes between 60° and 40°S, opal production is limited by silica supply and therefore, the increase in calcium carbonate production is stronger than the increase in opal production. Hence, towards the end of the 22nd century, the sea–air DMS-fluxes in this region increase stronger than indicated by PP. These results are consistent with the model study by Bopp et al. (2003) who also find increased DMS-production due to shifts from diatom to non-diatom species in the northern part of the Southern Ocean.

The global total marine DMS-emissions to the atmosphere are 24.2 Tg S yr$^{-1}$ in the control simulation, and average fluxes for the time period 2071 to 2100 decrease by 0.97 Tg S yr$^{-1}$ in our BASE RCP8.5 simulation relative to the control run (Table 2). The increasing sea–air fluxes in the Southern Ocean and the Arctic Ocean outweigh the decreases elsewhere towards the end of the 22nd century, and we find DMS-emissions that are globally slightly larger (0.12 Tg S yr$^{-1}$ for 2171 to 2200) in RCP8.5 relative to the control simulation.

## 3.2 Experiments SMED and SHIGH

The average pH of the surface ocean declines from 8.16 in the control simulation to 7.72 (average over 2071 to 2100) and 7.43 (2171 to 2200) under the RCP8.5 scenario and its extension (Fig. 4). If we, following Six et al. (2013), introduce a sensitivity of DMS-production to pH in our model as described above, global DMS-emissions in 2071 to 2100 are reduced relative to experiment BASE by 17% (4 Tg S yr$^{-1}$) and 27% (6.3 Tg S yr$^{-1}$) in the simulations SMED and SHIGH, respectively. Towards the end of the 22nd century (2171 to 2200), we find reductions of 31% (7.3 Tg S yr$^{-1}$) and 48% (11.5 Tg S yr$^{-1}$). The most pronounced reduction in absolute sea–air DMS-fluxes is found over the southern hemisphere (Fig. 5a to d), particularly between 40° and 50°S, where our model simulates the largest DMS-emissions in the control run.

This drastic reduction of sulfur input to the atmosphere leads to a reduction of SO$_2$ and sulfate aerosol loads. Since sulfate particles usually have a longer atmospheric lifetime than DMS, particularly in the pristine atmosphere over remote ocean regions, the pattern of changes in sulfate aerosol mass is regionally much smoother than the underlying sea–air DMS-flux changes (Fig. 5e to h). There is a pronounced north-south gradient of sulfate load reduction, with the largest reductions found in the south, despite the fact that the decrease in pH is largest in the Arctic Ocean (Fig. 4). In the Arctic DMS sea–air fluxes remain relatively small in experiment BASE even towards the end of the 22nd century. Therefore, although the percentage reduction of fluxes (per m$^2$ of ocean area) in SMED and SHIGH relative to BASE are largest in the Arctic Ocean, the decrease of absolute fluxes (integrated over the whole region) is much smaller than in the Southern Ocean. As a result, the absolute change in SO$_4$ column burden is three times larger over the Southern Ocean than over the Arctic region.

Close to strong natural (volcanic) and anthropogenic sulfur sources, we find a non-linear relationship between DMS-emissions and sulfate, e.g. sulfate loads are slightly increased rather than reduced over the northern Indian Ocean and adjacent land regions. Owing to the reduced sulfate mass in aerosol particles and a reduced aerosol number concentration (Fig. 6a to

c), we find a reduction in cloud droplet number concentrations (CDNC, Fig. 5i to l) over most ocean regions. Increases in CDNC are confined to land areas and to the Southern Ocean south of $60°$ S. Figure 6 demonstrates that the reduction of aerosol numbers and CDNC is found throughout the lower troposphere, being accompanied by an increase in the effective radius of liquid cloud droplets ($R_{\text{eff}}$).

Altered aerosol and cloud properties cause changes in the radiative balance of the atmosphere. Here, we investigate changes in the radiation balance at the top of our model (2.2 hPa). Changes in the short-wave net radiative flux ($N_s$) are the sum of changes in clear-sky net short-wave fluxes ($N_{s,c}$) due to altered aerosol scattering and absorption (direct aerosol effect) and changes in the short-wave cloud radiative effect (CRE$_s$), $\Delta N_s = \Delta N_{s,c} + \Delta \text{CRE}_s$ (with all fluxes scaled to the total grid-cell area). The cloud radiative effect (CRE) is evaluated in the model by parallel calls to the radiation code with and without clouds.

Differences in CRE between our transient simulations include contributions from changes in cloud-albedo, cloud-lifetime (first and second indirect effect), and from changes due to altered radiative heating by absorbing aerosols (semi-direct effect). Both, $\Delta N_{s,c}$ and $\Delta \text{CRE}_s$, also have contributions due to changes in surface albedo in our transient simulations (additional sea-ice melt, see Sect. 3.3). The analysis of changes in the long-wave radiation balance is complicated by the fact that the outgoing long-wave fluxes are altered due to different climate conditions in our experiments. The long-wave cloud radiative effect

(CRE$_l$) is available as model output, but our experiment design does not allow to separate the direct aerosol effect on outgoing long-wave radiation from other changes in the long-wave band. However, we can assume that the long-wave direct effect is generally negligible, since the diameter of DMS-derived aerosols is small compared to the long-wave radiation's wave-length.

We find that changes in the radiative fluxes depend approximately linearly on the sea–air DMS-flux anomaly, $\Delta F_{\text{DMS}}$, in our model (Fig. 7). Although there is considerable interannual variability, linear trends are virtually identical whether we use

data from the experiment SMED or SHIGH or use the data of both experiments combined. Clouds exert an average additional short-wave radiative effect of $-0.055$ W m$^{-2}$ (Tg S yr$^{-1}$)$^{-1}$. This effect is larger over the oceans than over land, where we also find a larger interannual variability (Fig. 7a to c). Changes in CRE$_l$ in our experiments are close to zero (Fig. 7d to f) and not discussed further. The reduced load of sulfate aerosols considerably alters $N_{s,c}$ by $-0.047$ W m$^{-2}$ (Tg S yr$^{-1}$)$^{-1}$, and again $\Delta N_{s,c}$ is significantly larger over the oceans than over land (Fig. 7g to i, see further discussion below). The global

average transient change of the total net radiative flux at the top of our model ($N_t$) is $-0.026$ W m$^{-2}$ (Tg S yr$^{-1}$)$^{-1}$. This includes a contribution with a positive trend over land ($+0.037$ W m$^{-2}$ (Tg S yr$^{-1}$)$^{-1}$), and a stronger negative contribution over the oceans ($-0.055$ W m$^{-2}$ (Tg S yr$^{-1}$)$^{-1}$, see Fig. 7j to l). The resulting negative radiative balance over land is due to the increased outgoing long-wave radiation at higher surface temperatures, which outweighs the weak changes in CRE over land. Thus, part of the excess-energy gained through the reduced DMS-load over the ocean is radiated back to space over land. The

global surface temperature anomaly is $-0.041$ K (Tg S yr$^{-1}$)$^{-1}$ in our experiments (Fig. 7m to o).

To set these results into perspective, we calculate the changes in radiative fluxes and in $T_s$ towards the end of the 22nd century using the average $\Delta F_{\text{DMS}}$ over 2171 to 2200 for SMED ($-7.3$ Tg S yr$^{-1}$) and SHIGH ($-11.5$ Tg S yr$^{-1}$). We also break our results down into three broad latitude bands: northern high latitudes (NHL, north of $40°$N), low latitudes (LL, $40°$S to $40°$N), and southern high latitudes (SHL, south of $40°$S). For the southern high latitudes, we define an additional region

SHLni, which excludes grid points that are covered by sea-ice in the control run. The average changes in CRE$_s$ at the end of

the 22nd century for experiment SMED are $0.40\,\mathrm{W\,m^{-2}}$ for the global domain, $0.50\,\mathrm{W\,m^{-2}}$ over the ocean, and $0.18\,\mathrm{W\,m^{-2}}$ over land. The corresponding values for the simulation SHIGH are 0.63, 0.78, and $0.28\,\mathrm{W\,m^{-2}}$ (Fig. 8a and b). The overall smaller cloud radiative effect over land is due to negative contributions from northern and southern high latitude land areas. We note that the relatively small $\Delta\mathrm{CRE}_s$ for the Southern Ocean (despite the strongest reduction of DMS-emissions) is due to the

5 additional melt of sea-ice around Antarctica in SMED and SHIGH. This feedback is further discussed in the next section. The effect of additional sea-ice melt is also seen in changes of the short-wave clear-sky radiation balance, $\Delta N_{s,c}$, for the Southern Ocean (Fig. 8c and d). The very high values of about $1\,\mathrm{W\,m^{-2}}$ for SMED and about $1.5\,\mathrm{W\,m^{-2}}$ for SHIGH are due to less reflected incoming solar radiation where sea-ice melts, but also due to the fact that modelled reductions in sulfate load are particularly high for the Southern Ocean.

The total transient changes in net radiative fluxes, $\Delta N_t$, in our experiments (Fig. 8e and f) are smaller than the changes of short-wave fluxes, since they includes contributions of increased long-wave radiation due to increased temperatures. We find values of 0.19, 0.40, and $-0.27\,\mathrm{W\,m^{-2}}$ (SMED), and 0.30, 0.63, and $-0.43\,\mathrm{W\,m^{-2}}$ (SHIGH) for the global domain, over the ocean, and over land, respectively. Note that $\Delta N_t$ is much larger south of $40^\circ$S than north of $40^\circ$N (e.g. 0.68 versus $0.39\,\mathrm{W\,m^{-2}}$ in SHIGH) due to the much smaller land area and a stronger reduction of DMS-emissions. Consequently, the

additional surface warming $\Delta T_s$ shows a pronounced asymmetry between northern and southern high latitudes. South of $40^\circ$S, we find $\Delta T_s$ values as high as 0.46 K (SMED) and 0.73 K (SHIGH) while the global average additional warming is 0.30 K (SMED) and 0.47 K (SHIGH). The largest additional warming in our simulations is found over the Antarctic continent (0.56 K SMED, and 0.89 K SHIGH), which is almost twice the global average.

The study of Six et al. (2013) uses a very similar version of the ocean biogeochemistry model HAMOCC and the same

parameterisations of DMS-production, pH-sensitivity, and DMS-emissions. However, compared to RCP8.5, the SRES A1B scenario used by these authors is much more moderate; the $CO_2$ concentration in this scenario reaches about 700 ppm in 2100.

If we, for comparison, consider the period 2046 to 2075, during which the average atmospheric $CO_2$ in our scenario simulations is 700 ppm, we find a reduction of DMS-emissions of 2.9 (SMED) and $4.4\,\mathrm{Tg\,S\,yr^{-1}}$ (SHIGH), which is very similar to the values presented in Six et al. (3.2 and $5\,\mathrm{Tg\,S\,yr^{-1}}$, respectively). Their estimate of the additional effective radiative forcing

($0.4\,\mathrm{W\,m^{-2}}$ for SMED and $0.64\,\mathrm{W\,m^{-2}}$ for SHIGH in 2100) has been derived from one-year simulations using a stand-alone atmospheric circulation model with aerosol chemistry, and does not include slow feedbacks. Therefore, the corresponding (transient) radiative imbalance of 0.08 and $0.12\,\mathrm{W\,m^{-2}}$ that we find in our SMED and SHIGH simulations for 2046 to 2075 is not comparable. If we calculate the expected equilibrium warming for NorESM using the Six et al. (2013) radiative forcing values with the equilibrium climate sensitivity for NorESM derived by Iversen et al. (2013, 2.87 K at $2\times CO_2$), we arrive at

0.31 (SMED) and 0.50 K (SHIGH). The additional transient surface warming of 0.12 and 0.18 K that we find for the period 2046 to 2075 in our simulations indicates that about 40% of this equilibrium warming has been realised, which seems to be reasonable in view of the rapid changes in DMS-fluxes ongoing in our simulations.

### 3.3 Feedbacks

#### 3.3.1 Carbon cycle feedbacks

We ran our experiments with freely evolving atmospheric $CO_2$, that is, atmospheric $CO_2$ concentrations are determined by anthropogenic emissions (including land use change emissions) and exchange with the land biosphere and the ocean. Hence, changes in climate, for example the additional surface warming due to reduced DMS-emissions, can be amplified by carbon cycle feedbacks in our model simulations. This is indeed the case, but the effect is rather small. By the end of the 22nd century, the ocean and land have released an additional amount of about 13 (26) Pg C in the experiments SMED (SHIGH) relative to BASE. This results in an atmospheric $CO_2$ concentration that is about 6 (12) ppm larger in SMED (SHIGH) than in BASE. Given that the atmospheric $CO_2$ concentration exceeds 2000 ppm at this point of time, this small feedback can be neglected.

#### 3.3.2 Sea-ice feedback

The additional warming in the experiments SMED and SHIGH leads to a reduced sea-ice extent relative to BASE. We discuss the related feedbacks for the last 30 years of the 22nd century since the additional melt is largest for this period. At this time, southern hemisphere annual average sea-ice extent has decreased from about $14.5 \times 10^6$ km$^2$ in the control simulation to $2.7 \times 10^6$ km$^2$ in experiment BASE. The additional decrease in SMED and SHIGH is $0.54 \times 10^6$ and $0.62 \times 10^6$ km$^2$. In contrast, in the Arctic Ocean, sea-ice cover decreases from $11.5 \times 10^6$ km$^2$ to only $0.23 \times 10^6$ km$^2$ in simulation BASE. Hence, as the northern hemisphere is virtually sea-ice free year round towards the end of the 22nd century, we concentrate our analysis on the Southern Ocean.

The reduced sea-ice area in SMED and SHIGH has a significant effect on the short-wave energy balance under clear sky conditions (Fig. 8c and d). If we consider sea-ice-free regions only (SHLni), we find that $\Delta N_{s,c}$ is 0.65 W m$^{-2}$ (SMED) and 1.02 W m$^{-2}$ (SHIGH) smaller than for the total ocean area south of 40°S (SHL). While this effect is straightforward to understand (more absorbed short-wave radiation due to less sea-ice), the short-wave radiative effect due to clouds shows the opposite behavior (Fig. 8a and b). We find increases in CRE$_s$ of 0.56 W m$^{-2}$ (SMED) and 0.89 W m$^{-2}$ (SHIGH) if only sea-ice-free grid points are considered. This can be explained by the fact that clouds have only a weak short-wave effect over ice surfaces in our model (clouds reflects incident radiation marginally better than ice surfaces). If the sea-ice is removed and replaced by open ocean, clouds have a much stronger negative radiative effect, since their reflectance increased relative to the underlying surface. The total effect of additional sea ice melt on the transient radiative imbalance over the ocean south of 40°S is small compared to the two opposing contributions (Fig. 8e and f). We find that $N_t$ increases by about 0.08 W m$^{-2}$ (SMED) and 0.13 W m$^{-2}$ (SHIGH) through the additional melt of sea-ice.

#### 3.3.3 DMS-climate feedbacks

Climate change caused by the pH-sensitivity of DMS-production also feeds back on the DMS-production itself. Although there is no significant difference in globally integrated PP between the runs BASE, SMED, and SHIGH, we find compensating re-

gional differences. The additional warming and increased incident short-wave radiation lead to an increase of PP south of $40°$S in SMED and SHIGH compared to BASE, while, at low latitudes and north of $40°$N, a reduction of PP in response to stronger stratification and less nutrient supply is found. We estimate the effect of these changes on DMS-production by calculating $P^*_{\mathrm{DMS}}$, which is the DMS-production based on $CaCO_3$ and silica export fields from the simulation BASE, combined with tem-

perature and pH-fields (and the pH-dependency) from the runs SMED or SHIGH. Hence, $P^*_{\mathrm{DMS}}$ is the DMS-production that would arise in the simulations SMED and SHIGH if there was no climate change relative to the simulation BASE. This defini-tion also includes differences in DMS-production due to changes in sea-ice cover. We then define the DMS-climate feedback on DMS-production as $P^f_{\mathrm{DMS}} = P_{\mathrm{DMS}} - P^*_{\mathrm{DMS}}$.

For our calculation of $P^*_{\mathrm{DMS}}$, we use monthly averaged model output. This introduces a systematic bias, since there is a causal

connection between pH- and PP-anomalies, and these anomalies occur on a sub-monthly time-scale during bloom-periods (PP increases pH during strong phytoplankton blooms due to the draw-down of dissolved inorganic carbon). By re-constructing the known DMS-production for SHIGH from monthly mean model output, we find that our calculations could underestimate the true value of $P^*_{\mathrm{DMS}}$ at high latitudes by up to $40\,\mathrm{mg\,S\,m^{-2}\,yr^{-1}}$ locally (with an average of about $10\,\mathrm{mg\,S\,m^{-2}\,yr^{-1}}$ south of $40°$S, where the effect is largest). Nevertheless, the underestimation of $P^*_{\mathrm{DMS}}$ (or equivalently, the overestimation of $P^f_{\mathrm{DMS}}$) is

smaller than 3% for the total production south of $40°$S.

We find that the DMS-climate feedback $P^f_{\mathrm{DMS}}$ is generally small, except south of $40°$S, where $P^f_{\mathrm{DMS}}$ locally exceeds $200\,\mathrm{mg\,S\,m^{-2}\,yr^{-1}}$ for simulation SHIGH (Fig. 9). This feedback is negative, that is, DMS-production increases with climate change, which counteracts the decrease due to acidification. The feedback accounts for more than 15% of the DMS-production almost everywhere south of $40°$S (Fig. 9d).

We estimate the DMS-climate feedback on DMS-fluxes, $F^f_{\mathrm{DMS}}$, by assuming that the fraction of the DMS-production re-leased as DMS to the atmosphere does not change between our simulations and the (hypothetical) simulation with the DMS-climate feedback excluded. By making this assumption, we neglect the fact that the fraction of DMS-production released to the atmosphere generally increases with decreasing DMS-production in our simulations (since the bacterial consumption de-creases). The effect on the allocation of the DMS-flux is small for the small differences between $P_{\mathrm{DMS}}$ and $P^*_{\mathrm{DMS}}$ discussed

here. However, we might tend to overestimate the feedback $F^f_{\mathrm{DMS}}$ by this assumption.

Globally, the increase of DMS-emissions due to the DMS-climate feedback in the Southern Ocean is nearly cancelled out by decreased fluxes at low latitudes and north of $40°$N (Fig. 10). Towards the end of the 22nd century (2171 to 2200) the feedback south of $40°$S accounts for 8.9% ($0.49\,\mathrm{Tg\,S\,yr^{-1}}$) in SMED and 15.5% ($0.61\,\mathrm{Tg\,S\,yr^{-1}}$) in SHIGH (see Table 3 for more details).

As mentioned in Sect. 3.1, the sensitivity of PP (and hence detritus and DMS-production) to climate change in the Southern Ocean seems to be high in our model compared to other models. Therefore, and because of the issues discussed above (use of monthly mean output fields, assumption of constant sea–air flux fraction) the negative DMS-climate feedback found in our simulations south of $40°$S might represent an upper estimate.

# 4 Summary and conclusions

We have simulated the impact of a possible reduction of marine DMS-emissions due to ocean acidification in a fully coupled Earth system model. The atmospheric model employs a state-of-the-art aerosol module that reacts to changes in atmospheric DMS-concentrations by changing the radiative balance of the Earth due to direct and indirect aerosol effects. All processes included in the ocean model's DMS-scheme (DMS-production as well as loss through bacterial consumption, photolysis, and sea–air fluxes) are affected by climate change due to their explicit dependence on environmental factors. Hence, we simulate the interplay between DMS-production and emissions, and the radiation balance of the Earth in a fully interactive fashion. Our set of model experiments consists of RCP8.5 scenario simulations that assume no (BASE), medium (SMED), and high (SHIGH) sensitivity of DMS-production to seawater pH.

We find a linear relationship between reduction of DMS sea–air fluxes and changes in radiative fluxes, as well as changes in surface temperature. Our study is consistent with the results of a previous off-line study and confirms the order of magnitude of the additional warming found. The global average transient warming in our simulation BASE (8.1 K towards the end of the 22nd century) is amplified by 3.7% (5.8%) in experiment SMED (SHIGH) through the pH-dependency of DMS-fluxes. However, we find that the additional surface warming is not spatially homogeneous, but has a strong north-south gradient with much stronger surface warming in the southern hemisphere. The additional transient warming over the Antarctic continent of 0.56 K (0.89 K) is almost twice the global average of 0.30 K (0.47 K).

Since our model simulates large reductions in sea–air DMS-fluxes (up to 48% in experiment SHIGH towards the end of the 22nd century), a considerable negative feedback loop should, according to the CLAW-hypothesis, counteract the original reduction. We indeed find a CLAW-like feedback in the Southern Ocean where the additional warming stimulates additional primary and DMS-production. We estimate that south of 40°S DMS-emissions would be further reduced by up to 15% in a model run excluding this feedback. North of 40°S, however, the additional warming tends to additionally weaken (or to only slightly enhance) DMS-fluxes, such that, at the global scale, there is no significant CLAW-like feedback in our simulations.

We have chosen a high emission scenario for our model experiments, and we assume a medium to high sensitivity of DMS-production to pH. Therefore, our results likely represent an upper limit of additional transient warming that might arise by this mechanism due to ocean acidification. If humankind decides to follow a low emission pathway, and consequently, the global ocean experiences much less severe acidification than projected in this study, the global additional temperature change might appear less significant. However, our study highlights that the additional warming due to a reduced DMS-production could reach almost twice the global average over the Antarctic region, which might have implications for the atmospheric $CO_2$-level that is allowable to avoid a destabilization of Antarctic ice shelves.

We finally note that the parameterisation for the pH-dependence of DMS-production is solely based on empirical evidence. A better mechanistic understanding of the processes leading to changes of DMS-production on the plankton community level is needed to better assess the climatic implications that ocean-acidification might have through altering the global sulfur cycle.

## 5  Code availability

The NorESM source code is available for scientific and educational purposes. As major model components are based on software developed by others, whose interests have to be protected, availability of the code is subject to signing a license agreement. Please mail to noresm-ncc@met.no for inquiries about obtaining the source code of NorESM. For the use of HAMOCC signing of the MPI-ESM license agreement is required in addition, which can be easily done through

http://www.mpimet.mpg.de/en/science/models/model-distribution.html

## 6  Data availability

The model data used in this study are long-term archived at the Norwegian Storage Infrastructure project (NorStore) for at least 5 years following publication and will be made available upon request.

*Competing interests.*  The authors declare that they have no conflict of interest.

*Acknowledgements.*  We thank David Archer and an anonymous reviewer for their helpful and constructive comments, which improved this manuscript. JS, JT, NG, AK, and ØS were supported by the Research Council of Norway through project EVA (229771). Supercomputer time and storage resources were provided by the Norwegian metacenter for computational science (NOTUR, project nn2345k), and the Norwegian Storage Infrastructure (NorStore, project ns2345k). This work was supported by the Bjerknes Centre for Climate Research. We gratefully acknowledge the CESM project, which is supported by the National Science Foundation and the Office of Science (BER) of the U.S. Department of Energy.

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

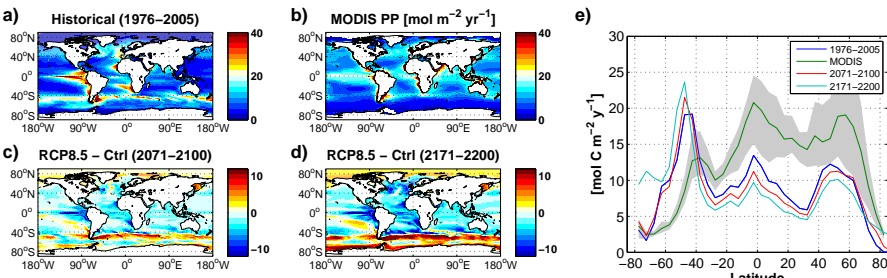

**Figure 1.** Vertically integrated annual primary production (PP, mol C m$^{-2}$ yr$^{-1}$) **(a)** in the experiment BASE (historical simulation, average over 1976 to 2005), **(b)** mean of three satellite based climatologies (derived from MODIS retrievals), **(c)** change found in the RCP8.5 scenario simulation relative to the control run towards the end of the 21st century (2071 to 2100), and **(d)** same as (c) but towards the end of the 22nd century (2171 to 2200). Panel **(e)** displays the zonal means of PP in the BASE historical simulation (dark blue) and for the RCP8.5 scenario (red, light blue). The green line with the grey shaded area represents the mean and range of the zonal means of the three satellite based climatologies.

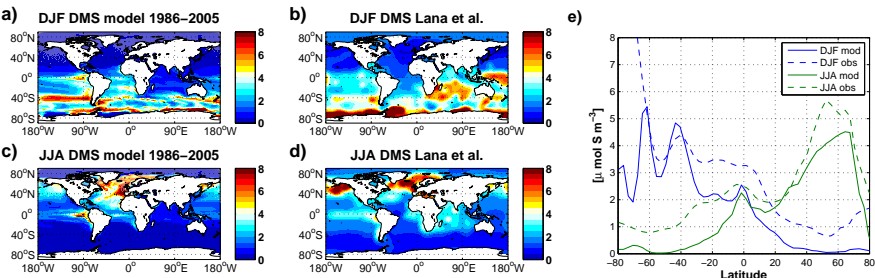

**Figure 2.** Surface DMS concentration ($\mu$mol S m$^{-3}$) during **(a, b)** boreal winter (DJF) and **(c, d)** boreal summer (JJA). Results for the BASE historical simulation averaged over 1986 to 2005 are shown in panels **(a, c)** and the observation-based climatology by Lana et al. (2011) is displayed in panels **(b, d)**. Panel **(e)** shows the zonal means of each field presented in panels **(a–d)**.

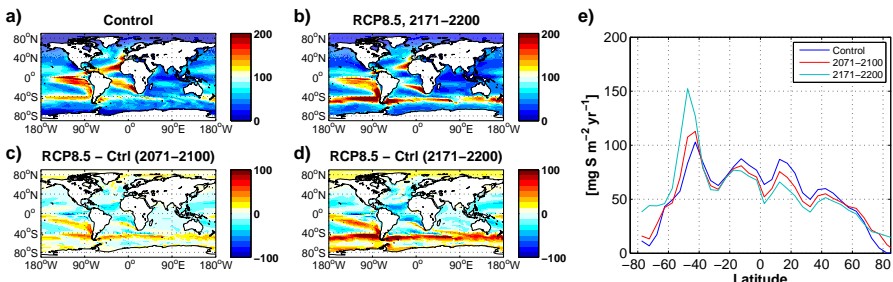

**Figure 3.** Modelled sea–air DMS-fluxes (mg S m$^{-2}$ yr$^{-1}$) in **(a)** the control run, and **(b)** the RCP8.5 scenario simulation BASE (no sensitivity of DMS-production to pH) towards the end of the 22nd century (2171 to 2200). Panels **(c)** and **(d)** display the change in sea–air DMS-fluxes in the RCP8.5 scenario BASE relative to the control simulation towards the end of the 21st (2071 to 2100) and 22nd (2171 to 2200) century, respectively. Panel **(e)** shows the zonal mean sea–air DMS-fluxes in the control and RCP8.5 simulations.

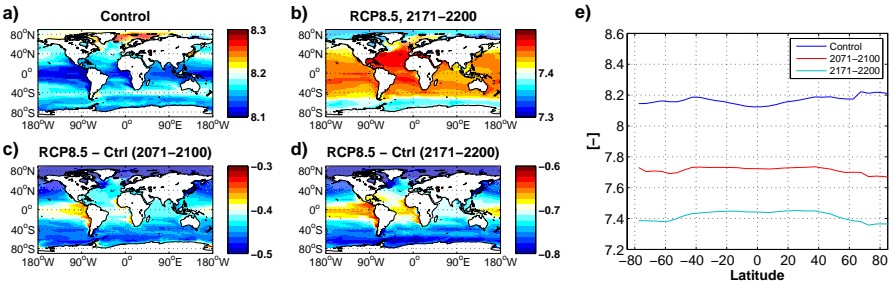

**Figure 4.** Surface ocean pH **(a)** in the control simulation, **(b)** in the RCP8.5 scenario simulation BASE (very similar for SMED and SHIGH) towards the end of the 22nd century (2171 to 2200), **(c)** change found in the RCP8.5 scenario simulation BASE relative to the control run towards the end of the 21st century (2071 to 2100), and **(d)** same as (c) but towards the end of the 22nd century (2171 to 2200). Panel **(e)** displays the zonal means of pH in the control run (dark blue) and for the RCP8.5 scenario BASE (red, light blue).

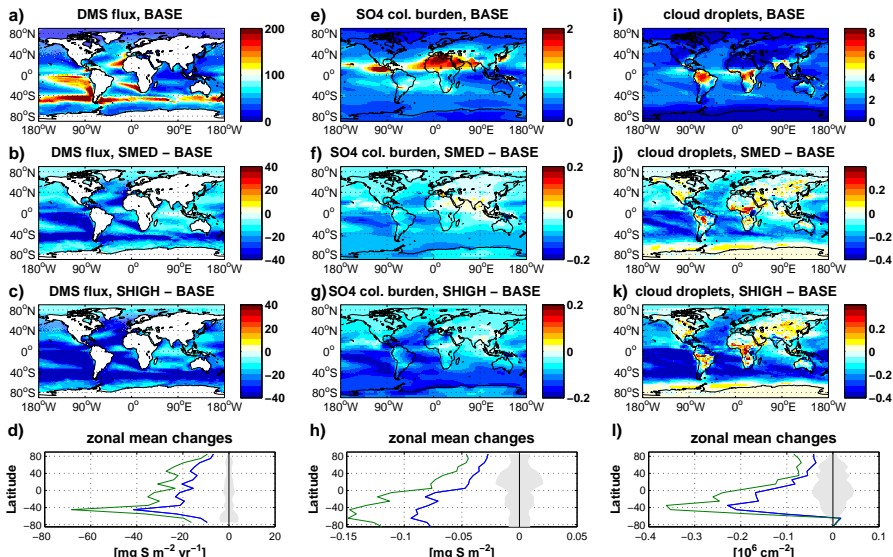

**Figure 5.** **(a)** Modelled sea–air DMS-flux (mg S m$^{-2}$ yr$^{-1}$) for scenario BASE (no sensitivity of DMS-production to pH), and **(b, c)** changes of sea–air DMS-fluxes relative to BASE in experiments SMED ($\gamma = 0.58$) and SHIGH ($\gamma = 0.87$), respectively. Panel **(d)** shows the zonal mean changes of sea–air DMS-fluxes for SMED (blue) and SHIGH (green), and the grey shaded area displays a measure for natural variability (standard deviation over 100 years of the control run). Panels **(e to h)** and **(i to l)** show corresponding plots for atmospheric sulfate column burden (mg S m$^{-2}$) and vertically integrated cloud droplet number concentration ($10^6$ cm$^{-2}$). All panels show results averaged over the period 2171 to 2200.

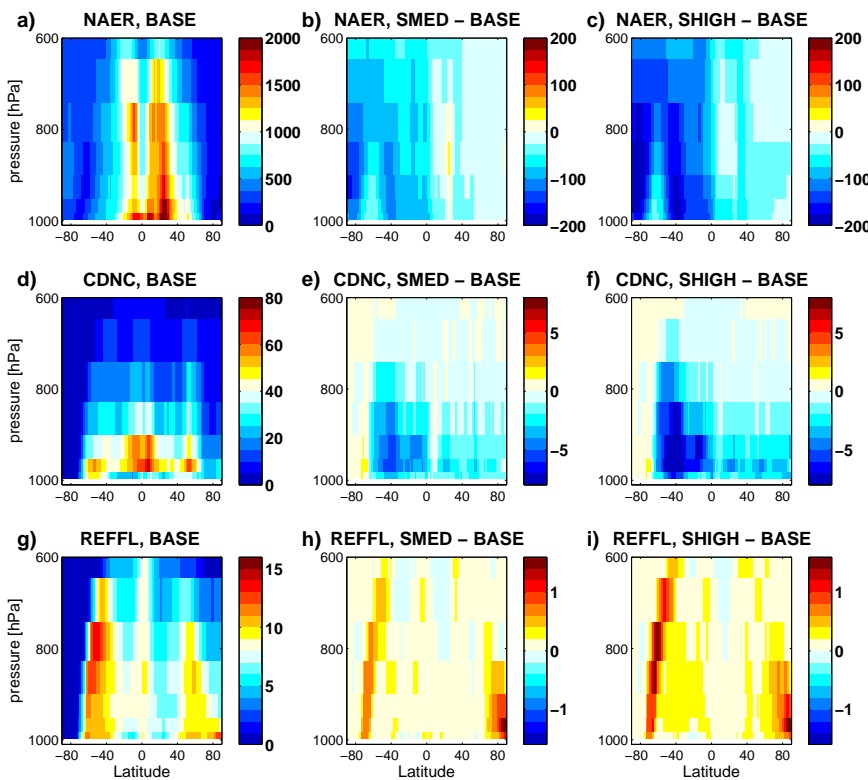

**Figure 6. (a)** Zonal mean aerosol number concentration ($N_{aer}$, cm$^{-3}$) averaged over the period 2171 to 2200 for the experiment BASE (no sensitivity of DMS-production to pH), and **(b, c)** changes of $N_{aer}$ in the experiments SMED and SHIGH (relative to BASE). Panels **(d to f)** and **(g to i)** display corresponding plots for cloud droplet number concentration (CDNC, cm$^{-3}$) and effective radius of liquid cloud droplets ($R_{eff}$, $\mu$m).

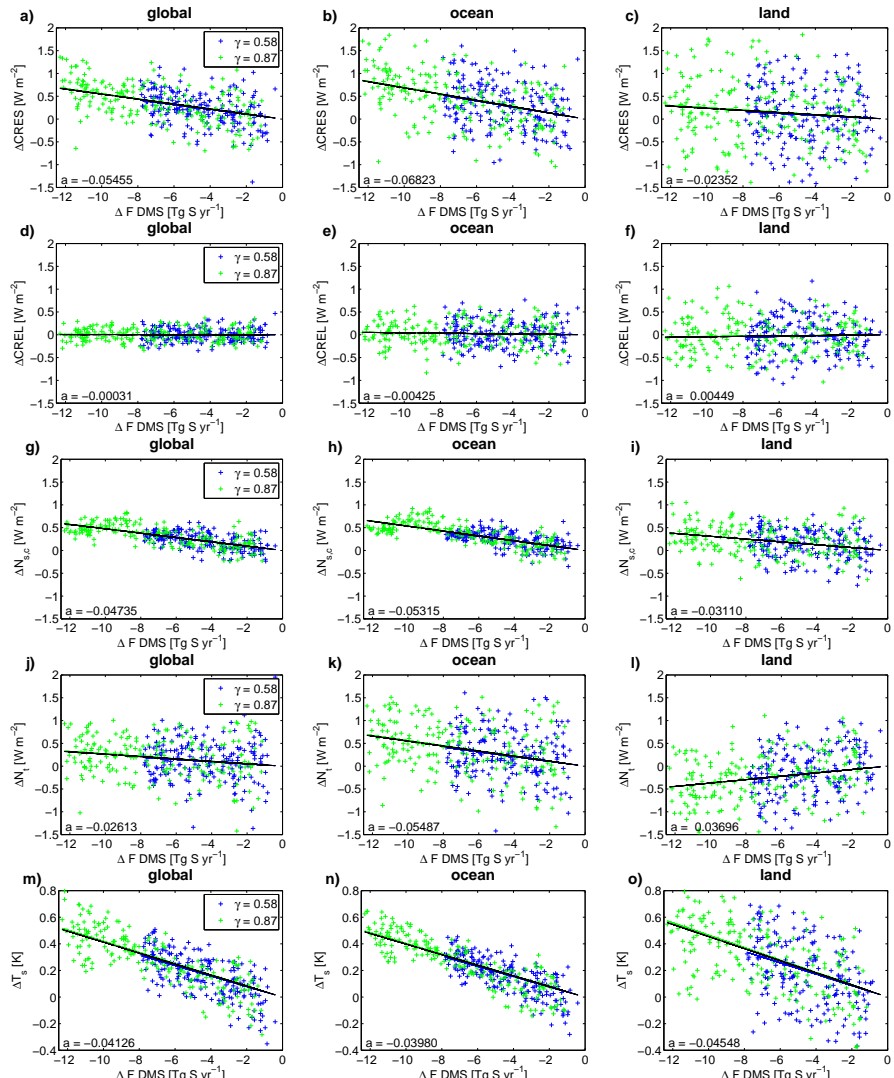

**Figure 7.** Scatter plots of anomalies of sea–air DMS-fluxes ($\Delta F_{\mathrm{DMS}}$, Tg S yr$^{-1}$) versus anomalies of **(a to c)** short-wave cloud radiative effect ($\Delta \mathrm{CRE}_s$, W m$^{-2}$), **(d to f)** long-wave cloud radiative effect ($\Delta \mathrm{CRE}_l$, W m$^{-2}$), **(g to i)** clear-sky net short-wave radiative flux at the top of model ($\Delta N_{s,c}$, W m$^{-2}$), **(j to l)** total radiative flux at the top of model ($\Delta N_t$, W m$^{-2}$), and **(m to o)** near surface temperature ($\Delta T_s$, K) for the experiments SMED (blue) and SHIGH (green) relative to experiment BASE. The first column of plots displays global annual averages for each year in 2006 to 2200, while for the second and third column the annual mean values were calculated for ocean and land grid-points, respectively. A linear fit to the data of each experiment (solid blue and green lines) as well as a linear fit to all data (solid black line) is shown in each panel. Note that in most cases the fit to individual experiments is indistinguishable from the fit to all data points and is hidden behind the black solid lines. The slope of the fit to all data is given in the lower left corner of each panel.

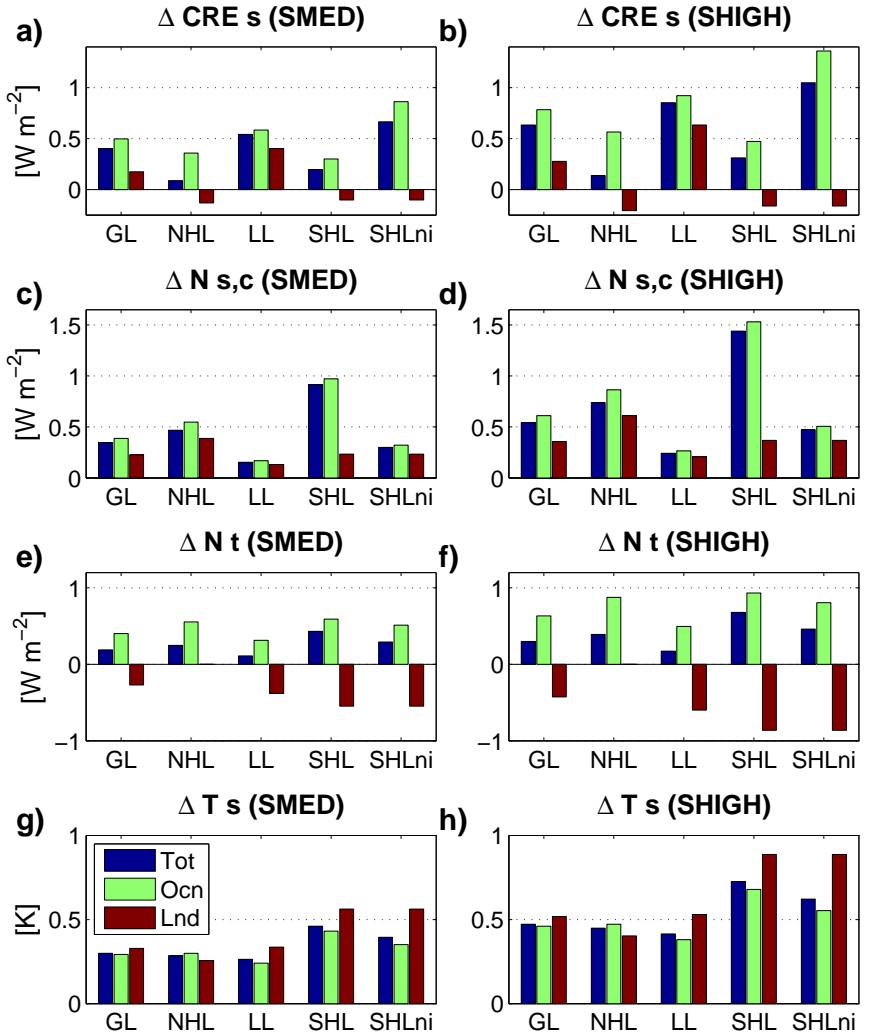

**Figure 8.** Changes of short-wave cloud radiative effect relative to experiment BASE ($\Delta\mathrm{CRE}_s$, $\mathrm{W\,m^{-2}}$) for the experiment **(a)** SMED and **(b)** SHIGH towards the end of the 22nd century (2171 to 2200). The groups of bars indicate averages over different latitude bands (GL: global; NHL: northern high latitudes, north of 40°N; LL: low latitudes, 40°S to 40°N; SHL: southern high latitudes, south of 40°S). For the southern high latitudes there is an additional group of bars (SHLni) that indicates averages over the area that is sea-ice free in the control run. Blue bars indicate the total area average, while green and brown bars indicate averages over ocean and land areas, respectively. **(c, d)** same as panels a and b, but for changes of the top-of-model short-wave net flux under clear sky conditions ($\Delta N_{s,c}$). **(e, f)** same as panels a and b, but for changes of the top-of-model total net radiative flux ($\Delta N_t$). **(g, h)** same as panels a and b, but for changes of surface temperature ($\Delta T_s$).

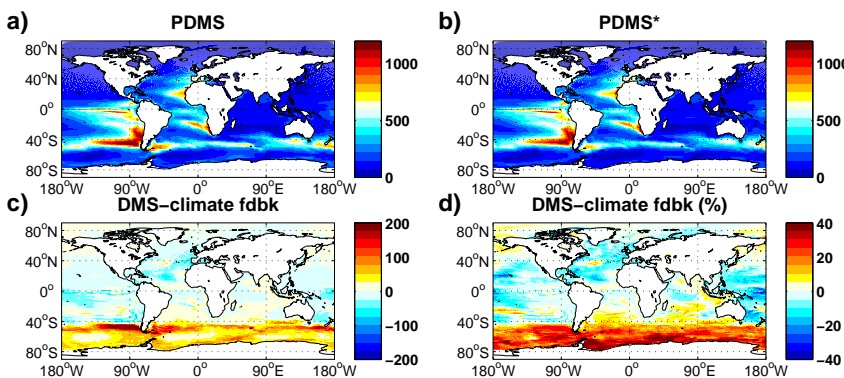

**Figure 9.** Averages over 2171 to 2200 of **(a)** DMS-production for the experiment SHIGH, **(b)** an estimate of the DMS-production that would occur in SHIGH without DMS-climate feedback ($P^*_{\mathrm{DMS}}$), **(c)** the corresponding estimate of the DMS-climate feedback $P^f_{\mathrm{DMS}}$ in absolute values, and **(d)** the DMS-climate feedback expressed in percent of the total DMS-production shown in panel (a). Units in (a to c) are $\mathrm{mg\,S\,m^{-2}\,yr^{-1}}$.

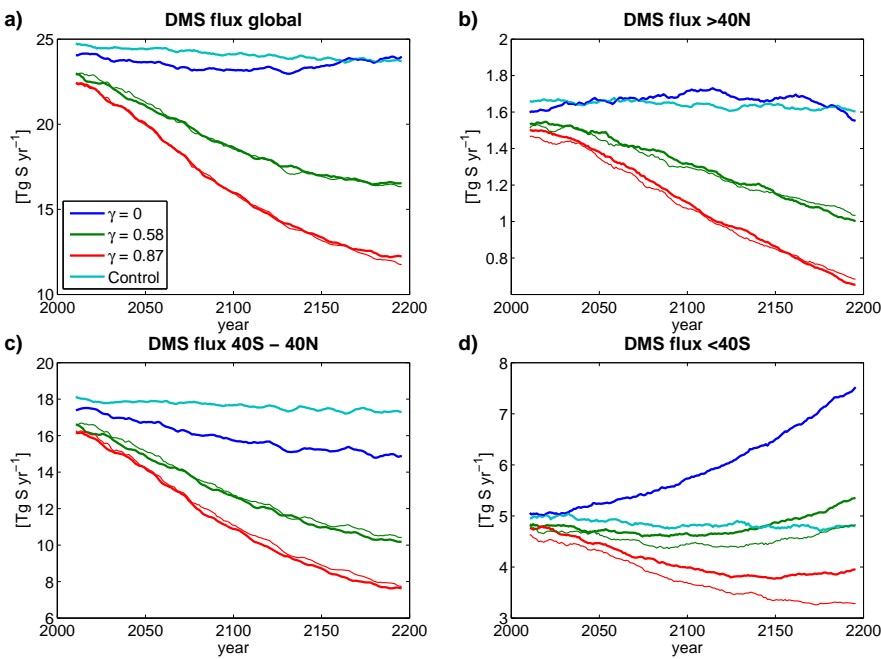

**Figure 10.** Time series of DMS-fluxes (smoothed by a 10-year running mean filter) for the RCP8.5 scenario and its extension to 2200 for the experiments BASE (dark blue lines), SMED (green) and SHIGH (red). The control run fluxes are indicated by the light blue lines. Thin green and red lines give an estimate of the fluxes that would occur without DMS-climate feedback ($F_{\mathrm{DMS}}^{*}$).

**Table 1.** Model experiments conducted for this study. All experiments have been run over the time period 1850–2200. The experiments BASE, SMED and SHIGH use prescribed $CO_2$-emissions (and other forcings) following the CMIP5 protocols for the historical simulation (1850–2005), the RCP8.5-scenario (2006–2100) and the RCP8.5-scenario extension (2101–2200).

| Experiment name | $CO_2$-emissions | pH-sensitivity |
|---|---|---|
| control | none | $\gamma = 0$ |
| BASE | historical/RCP8.5/RCP8.5-extension | $\gamma = 0$ |
| SMED | historical/RCP8.5/RCP8.5-extension | $\gamma = 0.58$ |
| SHIGH | historical/RCP8.5/RCP8.5-extension | $\gamma = 0.87$ |

**Table 2.** Global mean atmospheric $CO_2$ concentration, surface temperature ($T_s$), marine primary production (PP), and sea–air DMS-fluxes ($F_{\mathrm{DMS}}$) in the control run and changes in experiment BASE relative to the control run.

| | | Control 2071–2100 | BASE−Control 2071–2100 | BASE−Control 2171–2022 |
|---|---|---|---|---|
| $CO_2$ | (ppm) | 284 | $945 - 284$ | $1951 - 284$ |
| $T_s$ | (K) | 285.8 | 4.2 | 8.1 |
| PP | ($Pg\,C\,yr^{-1}$) | 44.2 | $-4.2$ | $-5.3$ |
| $F_{\mathrm{DMS}}$ | ($Tg\,S\,yr^{-1}$) | 24.2 | $-0.97$ | 0.12 |

**Table 3.** Quantification of the DMS-climate feedback on sea–air DMS-fluxes in our experiments (averages over the period 2171 to 2200). $F_{\mathrm{DMS}}$ denotes DMS-fluxes including all feedbacks, while $F_{\mathrm{DMS}}^{f}$ gives the estimated fraction of these fluxes that arise due to feedbacks (see Sect. 3.3.3).

| | | global | 40°N–90°N | 40°S–40°N | 90°S–40°S |
|---|---|---|---|---|---|
| **SMED** | | | | | |
| $F_{\mathrm{DMS}}$ | ($Tg\,S\,yr^{-1}$) | 16.56 | 1.04 | 10.31 | 5.21 |
| $F_{\mathrm{DMS}}^{f}$ | ($Tg\,S\,yr^{-1}$) | 0.10 | -0.03 | -0.27 | 0.46 |
| Fraction | (%) | 0.6 | -3.4 | -2.6 | 8.9 |
| **SHIGH** | | | | | |
| $F_{\mathrm{DMS}}$ | ($Tg\,S\,yr^{-1}$) | 12.35 | 0.70 | 7.76 | 3.89 |
| $F_{\mathrm{DMS}}^{f}$ | ($Tg\,S\,yr^{-1}$) | 0.30 | -0.03 | -0.21 | 0.61 |
| Fraction | (%) | 2.4 | -3.9 | -2.7 | 15.5 |