# Peer review of "Amplification of global warming through pH-dependence of DMS-production simulated with a fully coupled Earth system model"

_Biogeosciences, 2017_

## Referee Comment (RC1) · D. Archer (Referee) · 31 Mar 2017

This paper extends previous modeling simulations of the effect of changes in DMS production by phytoplankton caused by ocean acidification, and how these might or might not feed back to climate and further alter DMS production. A previous study had simulated the impact on DMS production in an off-line sense, with no mechanism for the feedback. I guess it's not a huge surprise that the climate impact of the DMS changes do not further derail DMS production very much (with exceptions noted in the paper). But the result is still useful as a mostly negative result, especially since it's documented well and estimated with state-of-the-art models. The presentation is very clear and polished; especially nice are the discussion of the CLAW hypothesis, and the

clear admission of the small scale of the feedbacks, rather than trying to spin them as positive to make them seem more exciting.

---

## Referee Comment (RC2) · Anonymous Referee #2 · 6 Apr 2017

Schwinger et al. present the results of a high emission scenario with a coupled Earth system model in which marine DMS-production has been linked to ocean acidification. A similar study (Six et al. 2013) has been already performed but without a fully coupled model system (the authors applied a two-step approach where the simulated marine emissions were used as an input for an atmospheric circulation model with aerosol chemistry). The advantage of using a fully coupled system is that feedbacks are enabled and their effects can be studied. Thus, the new work by Schwinger et al. is a necessary extension of the previous work, but unfortunately without exciting results and scientifically thrilling insights. (*But perhaps this cannot be expected...*).

In any case, I expect any scientific article to be self-contained. Unfortunately this is

not always the case, here. For instance, model assumptions are not clearly stated. In addition, several aspects remain unclear and need to be explained better.

- in several places (page 3, page 9) the authors mention that they use the **same** (page 3, line 1), **similar** (page 3, line 30) or **very similar** (page 9, line 24) biogeo-chemistry model as in the previous study by Six et al. (2013). These contradictory statements are vague and not very helpful. The differences and the motivation for them need to be spelled out and the consequences for model results need to be explained.

- page 4: as stated on page 2, line 3 "DMS is a by-product of marine primary production" (strictly speaking, it is DMSP). Nevertheless, the authors couple DMS production not to *primary* but to *export* production without further explanation. Primary and export production are uncoupled in most oceanic regions. Thus, the results must be significantly different if the more correct approach (which is relatively straightforward) is applied. Instead of using the exported calcite or opal matter to distinguish high and low DMS-producing phytoplankton, the degree of silicate limitation during phytoplankton growth could be taken. A more detailed explanation for this approach is needed.

- On a more fundamental level, I am puzzled about the idea behind implement-ing the relationship between DMS-production and pH in the model in the first place. As far as I understand the results of the different mesocosm studies, DMSP-production is even enhanced with decreasing pH but the DMS produc-tion decreased, most likely due to bacterial decomposition. I wonder whether the model *without the prescribed relationship* is able to reproduce some of this dynamic as DMS consumption is temperature and concentration-dependent? If so, it might be problematic that certain processes governing sources and sinks of DMS are included in the model but are now "overwritten" through prescribing such a relationship.

[Figure]

- page 4, line 26: the terms in the equation should be explained.

- page 5, line 21ff: why is pH-dependence only included from 2005 onwards?

- page 5, line 21ff: the experimental set-up must be explained much better. An overview table with details about all simulations would facilitate the understanding of the different model results. Also, what is the purpose of a model experiment from 1850-2200 with constant pre-industrial setting (control run)?

- page 6, line 10ff. and Fig. 1: why are the observations compared to the pre-industrial situation (control run) and not to the climatology from the historical simulation (e.g. 1975-2005)?

- page 7, line 6, line 16, ...: the authors often explain patterns in the DMS concentrations with those from primary production. However, DMS production has been coupled to *export* production in the model, thus DMS concentrations should be compared with patterns of export production. Something is inconsistent, here.

- It is unclear what insights can be deduced from the comparison of the BASE run (without pH-driven DMS changes) and the Control run (with pre-industrial setting).

- a figure showing the projected pH changes in the surface ocean would be helpful. In this context, I am wondering why the effects on DMS-fluxes are much stronger in the Southern Ocean compared to the Arctic Ocean. I would have expected a much stronger effect in the Arctic Ocean due to freshening and related higher drop in pH. Do the authors have an explanation why this is not the case?

- last sentence, page 12, line 31: is there really an agreement about "dangerous climate change"?

---

## Author Comment (AC1) · 24 May 2017

We would like to thank David Archer for reviewing our manuscript and for his positive comments.

---

## Author Comment (AC2) · 24 May 2017

**Authors' response to an anonymous review (reviewer #2) of "Amplification of global warming through pH-dependence of DMS-production simulated with a fully coupled Earth system model " by Schwinger et al.**

We thank the reviewer for reviewing our manuscript and for his/her constructive and helpful comments. Our detailed response to the points raised is given below (reviewer's comments in italic font, our response in normal font; page and line numbers refer to the submitted version of the manuscript).

I expect any scientific article to be self-contained. Unfortunately this is not always the case, here. For instance, model assumptions are not clearly stated. In addition, several aspects remain unclear and need to be explained better.

• in several places (page 3, page 9) the authors mention that they use the same (page 3, line 1), similar (page 3, line 30) or very similar (page 9, line 24) biogeochemistry model as in the previous study by Six et al. (2013). These contradictory statements are vague and not very helpful. The differences and the motivation for them need to be spelled out and the consequences for model results need to be explained.

We agree that we have not been clear enough here. There are small differences in the model versions used, e.g. the inorganic carbon chemistry is slightly different and some technical modifications were necessary for the implementation into the isopycnic MICOM model. Further, some parameters were differently tuned because of the different physical ocean models (details are described in Schwinger et al. 2016). For the purpose of this study, none of these changes is expected to be important. Further, the DMS-parametrisation as well as the parameterisation of pH-dependency are identical.

**Proposed changes to the manuscript:**

page 3, line 1: "The marine biogeochemistry model employed in this study is very similar to the one used by Six et al. (2013), with only minor differences (see Sect. 2.1). The prognostic DMS scheme as well as the assumptions on the pH-dependency of DMS-production are the same as in Six et al. (2013)."

page 3, line 30: "As a result, our model version is very similar but not identical to the one described by Ilyina et al. (2013). For example, the inorganic carbon chemistry has been updated for our model version and some technical modifications were necessary for the implementation into the isopycnic MICOM. Further, some parameters were differently tuned because of the different physical ocean models (see Schwinger et al. 2016 for details). None of these differences is expected to alter the findings and conclusions of this study."

page 9, line 24: "The study of Six et al. (2013) uses a very similar version of the ocean biogeochemistry model HAMOCC,..."

• page 4: as stated on page 2, line 3 "DMS is a by-product of marine primary production" (strictly speaking, it is DMSP). Nevertheless, the authors couple DMS production not to primary but to export production without further explanation. Primary and export production are uncoupled in most oceanic regions. Thus, the results must be significantly different if the more correct approach (which is relatively straightforward) is applied. Instead of using the exported calcite or opal matter to distinguish high and low DMS-producing phytoplankton, the degree of silicate limitation during phytoplankton growth could be taken. A more detailed explanation for this approach is needed.

We agree that this needs clarification. First, our statement that DMS-production is coupled to export production was actually not entirely correct. DMS-production is coupled to *detritus* production in our model. Since in HAMOCC, which is a relatively simple biogeochemistry model, these two are closely related (see below), we sloppily used "export" and "detritus" production interchangeably, which was confusing for the reader. Since the release of DMSP from phytoplankton cells is thought to happen mainly during senescence, viral attacks and grazing followed by a rapid conversion of DMSP to DMS by bacterial and algal enzymes (Stefels et al., 2007, and references therein), the coupling of DMS-production to detritus production is a simplifying but reasonable assumption.

Second, we acknowledge (also in view of the following comment made by the reviewer) that the description of the biogeochemistry model employed for this study was not comprehensive enough. HAMOCC is a relatively simple model: It only describes two functional types, phytoplankton and zooplankton. There are no separate functional types for calcifiers and silicifiers. Therefore, silicate limitation is not considered during phytoplankton growth, but is invoked only to determine the fraction of opal and  $CaCO_3$  in sinking shell material. Bacteria are not modeled explicitly, rather, processes related to bacterial activity (e.g. remineralisation of dissolved and particulate organic matter) are assumed to proceed at constant rates. The export fraction relative to PP is almost constant over the globe in this model (about 20% with the parameter settings used for this study) and is virtually equal to detritus production as mentioned above. Hence, for the HAMOCC model version employed in this study, patterns of DMS-production would not be different if it was linked to primary instead of detritus production.

**Proposed changes to the manuscript:**

We will replace all occurrences of "export production" with "detritus production" when used in the context of DMS-production. We further rewrite the HAMOCC description and the description of the DMS parameterisation as follows:

page 4, line 3ff: "[...] The model has only one generic phytoplankton and one generic zooplankton type. Bacteria are not modeled explicitly, rather, processes related to bacterial activity (e.g. remineralisation of dissolved and particulate organic matter) are assumed to proceed at constant rates. [...] Detritus is formed through grazing activity as well as phytoplankton and zooplankton mortality. Once

formed, it sinks through the water column at a constant speed of  $5 \text{ m d}^{-1}$  and is remineralised at a constant rate. Although calcifying and silicifying plankton functional types are not modeled explicitly, the fraction of calcium carbonate and biogenic silica that is added to the pool of sinking shell material depends on the availability of silicic acid. Here, it is implicitly assumed that diatoms out-compete other phytoplankton species when the supply of silicic acid is ample."

page 4, line 13ff: "[...] Observations indicate that when the cell membrane of phytoplankton is disrupted by senescence or due to viral attack and zooplankton grazing, the DMS-precursor dimethylsulphoniopropionate (DMSP) is released into sea water and rapidly converted to DMS by bacterial and algal enzymes (Stefels et al., 2007). Therefore, the DMS-production in our model is assumed to be a function of detritus production. It is further modified by the production of opal and CaCO3 shell material, that is, calcite or opal producing organisms are assumed to have different sulfur to carbon ratios. [...]"

• On a more fundamental level, I am puzzled about the idea behind implementing the relationship between DMS-production and pH in the model in the first place. As far as I understand the results of the different mesocosm studies, DMSP-production is even enhanced with decreasing pH but the DMS production decreased, most likely due to bacterial decomposition. I wonder whether the model without the prescribed relationship is able to reproduce some of this dynamic as DMS consumption is temperature and concentration-dependent? If so, it might be problematic that certain processes governing sources and sinks of DMS are included in the model but are now "overwritten" through prescribing such a relationship.

In HAMOCC none of the modeled processes is dependent on pH (except for the optional pH-dependency of DMS-production used in this study). pH is a diagnostic quantity in the model, which changes with the uptake of anthropogenic carbon and through biological processes as a consequence of inorganic carbon chemistry. Importantly, biological processes do not respond to changes in pH, e.g., there is no dependence of cell carbon quotas or stoichiometric ratios on inorganic carbon availability.

The model without the prescribed pH-dependency of DMS-production (our experiment BASE) shows shifts in DMS sea-air fluxes due to climate change. For example, primary and detritus production change in response to increased stratification at low latitudes, and the fraction of DMS-production that is consumed by bacteria increases with increasing temperature. All these shifts are, however, completely independent of pH. Therefore, when we apply the prescribed relationship between pH and DMS-production, we do not "overwrite" a process that the model would simulate anyway.

The fact that we introduce a decrease of DMS-production with decreasing pH (and not an increase in bacterial consumption) is defensible in view of available

knowledge and data. The role of bacteria in the DMS-cycle is at least twofold: They convert DMSP released from phytoplankton cells into DMS by enzymatic cleavage, which is thought to be the main DMS-production pathway, and they also metabolize DMS, which is the "bacterial consumption" in our model. (e.g., Stefels et al., 2007). Both processes might be pH dependent (Archer et al., 2013), but experimental data is scarce. Generally, shifts in species composition and grazing have been observed in mesocosm experiments under increasing  $[H^+]$  (Archer et al., 2013; Park et al., 2014), and are thought to be a major cause for decreases in DMS-production. Such shifts would introduce changes of the DMS-production term, not the bacterial consumption. We will explain these aspects better in a revised version of our manuscript.

**Proposed changes to the manuscript:**

page 4, add after line 10: "We note that HAMOCC's ecosystem model has no dependency on inorganic carbon availability or pH, e.g., neither cell carbon quotas or stoichiometric ratios nor calcification rates are assumed to vary with changing pH."

page 4, add after line 28: "We note that the reason for the observed decrease in DMS-concentrations under low pH is still debated. Shifts in species composition seem to play a major role in general (Archer et al., 2013; Park et al., 2014), but also changes in the rate of DMSP to DMS conversion and the rate of loss through bacterial consumption might contribute to the observed pH-dependency. All these processes cannot be resolved explicitly by our model, and, as a first approach, we assign a linear pH-dependency to DMS-production."

• page 4, line 26: the terms in the equation should be explained.

**Proposed changes to the manuscript:**

page 4, line 26ff: "Here, pH is the modeled local pH-value and pHpi is the local pH of the pre-industrial undisturbed ocean, relative to which a deviation of pH is measured. pHpi is taken from a monthly climatology calculated from 10 years of a preindustrial control simulation. The constant  $\gamma$  defines the sensitivity of DMS-production to pH, and is chosen following Six et. al (2013, see Sec. 2.3)."

• page 5, line 21ff: why is pH-dependence only included from 2005 onwards? page 5, line 21ff: the experimental set-up must be explained much better. An overview table with details about all simulations would facilitate the understanding of the different model results. Also, what is the purpose of a model experiment from 1850-2200 with constant pre-industrial setting (control run)?

We will include Table 1 (as shown at the bottom of this text) in a revised version of our manuscript. This will also clarify that pH-dependence is included over 1850-2200 for the SMED and SHIGH model experiments (not only from 2005 onwards).

The purpose of the control simulation is to account for model drift. This is to some degree already explained in the last paragraph of Section 2.3. We propose to make this point clearer by re-writing the beginning of this paragraph as follows.

**Proposed changes to the manuscript:**

page 5, line 31ff: "The control simulation is used to account for remaining model drift. Most importantly, there is a small decrease in primary production of about  $0.57 \,\mathrm{Pg}\,\mathrm{C\,century^{-1}}$ , and a corresponding decrease in sea-air DMS-fluxes of  $0.51 \,\mathrm{Tg}\,\mathrm{S\,century^{-1}}$ , due to the fact that fluxes of nutrients to the sediment are not replenished by any mechanism in the model version employed for this study."

• page 6, line 10ff. and Fig. 1: why are the observations compared to the preindustrial situation (control run) and not to the climatology from the historical simulation (e.g. 1975–2005)?

We agree that this is not very consistent (although the differences between the control run and the end of the historical period are not very large for PP). We will change Figure 1a and 1e to show a climatology over 1975–2005 (instead of a control run climatology) for a revised version of our manuscript.

• page 7, line 6, line 16, ...: the authors often explain patterns in the DMS concentrations with those from primary production. However, DMS production has been coupled to export production in the model, thus DMS concentrations should be compared with patterns of export production. Something is inconsistent, here.

As explained and discussed above (point 2), we will replace "export" with "detritus" production. One could introduce a new additional figure showing detritus production. However, on the annual time-scale considered here, detritus production very closely follows PP, and this figure would almost exactly be a scaled version of Figure 1 of the submitted manuscript (Figure 1 multiplied by 0.2). Further, PP can be compared to global satellite based estimates. Since there wouldn't be any additional insight gained by presenting a figure of detritus production, we instead propose to better explain the tight connection between PP and detritus production in a revised version of the manuscript as follows.

**Proposed changes to the manuscript:**

page 7, line 9ff: "The change of DMS-fluxes to the atmosphere in the RCP8.5 scenario simulation (Fig. 3) mainly follows the change in primary production. This is to be expected, since in our model detritus production very closely follows PP on an annual time-scale (about 20% with little spatial variability). However, [...]"

• It is unclear what insights can be deduced from the comparison of the BASE run

(without pH-driven DMS changes) and the Control run (with pre-industrial setting).

Changes of PP, detritus and DMS-production in the BASE simulations relative to the control run are the part of changes that are caused by climate change, without any contribution of ocean acidification. Modifications of DMS-production in SMED and SHIGH due to ocean acidification come on top of these changes. Hence, the BASE simulation is the background against which the changes due to ocean acidification have to be evaluated. We try to make this clearer by adding an introductory sentence to section 3.1.

**Proposed changes to the manuscript:**

page 6, add before line 4: "To begin with, we examine changes of PP and DMS sea-air fluxes in the BASE simulation relative to the pre-industrial control run. These changes are caused by climate change alone (no influence of ocean acidification). Additional changes relative to the BASE experiment that are caused by the pH-sensitivity of DMS-production in SMED and SHIGH will be analyzed and discussed in Sects. 3.2 and 3.3."

• a figure showing the projected pH changes in the surface ocean would be helpful. In this context, I am wondering why the effects on DMS-fluxes are much stronger in the Southern Ocean compared to the Arctic Ocean. I would have expected a much stronger effect in the Arctic Ocean due to freshening and related higher drop in pH. Do the authors have an explanation why this is not the case?

We will add a figure showing the projected pH changes to a revised version of the manuscript (included as Fig. 1 in this response below). The pH-changes in the Arctic Ocean are larger by roughly 0.1 compared to the tropics. Compared to the Southern Ocean, pH changes are still larger but only of the order of 0.03. There are two reasons why the changes in DMS-fluxes and the resulting climate effects are larger for the Southern Ocean. We will add an explanation for this in a revised version of our manuscript.

**Proposed changes to the manuscript:**

replace page 7, line 34ff by: "There is a pronounced north-south gradient of sulfate load reduction, with the largest reductions found in the south, despite the fact that the decrease in pH is largest in the Arctic Ocean (Fig. 4). In the Arctic DMS sea-air fluxes remain relatively small in experiment BASE even towards the end of the 22nd century. Therefore, although the percentage reduction of fluxes (per m2 of ocean area) in SMED and SHIGH relative to BASE are largest in the Arctic Ocean, the decrease of absolute fluxes (integrated over the whole region) is much smaller than in the Southern Ocean. As a result, the absolute change in SO4 column burden is three times larger over the Southern Ocean than over the Arctic region."

• last sentence, page 12, line 31: is there really an agreement about "dangerous climate change"?

Arguably, there is an agreement that a destabilization of Antarctic ice shelves would be "dangerous". We agree that it would be better to write this straightforward, and propose to reword this sentence.

**Proposed changes to the manuscript:**

"However, our study highlights that the additional warming due to a reduced DMSproduction could reach almost twice the global average over the Antarctic region, which might have implications for the atmospheric  $CO_2$ -level that is allowable to avoid a destabilization of Antarctic ice shelves."

Table 1: Model experiments conducted for this study. All experiments have been run over the time period 1850-2200. The experiments BASE, SMED and SHIGH use prescribed CO2-emissions (and other forcings) following the CMIP5 protocols for the historical simulation (1850-2005), the RCP8.5-scenario (2006-2100) and the RCP8.5-scenario extension (2101-2200).

| Experiment name | CO 2 -emissions         | pH-sensitivity  |
|-----------------|------------------------------------|-----------------|
| control         | no $CO_2$ emissions                | $\gamma = 0$    |
| BASE            | historical/RCP8.5/RCP8.5-extension | $\gamma = 0$    |
| SMED            | historical/RCP8.5/RCP8.5-extension | $\gamma = 0.58$ |
| SHIGH           | historical/RCP8.5/RCP8.5-extension | $\gamma = 0.87$ |

Figure 1: Surface ocean pH (a) in the control simulation, (b) in the RCP8.5 scenario simulation BASE (very similar for SMED and SHIGH) towards the end of the 22nd century (2171 to 2200), (c) change found in the RCP8.5 scenario simulation BASE relative to the control run towards the end of the 21st century (2071 to 2100), and (d) same as (c) but towards the end of the 22nd century (2171 to 2200). Panel (e) displays the zonal means of pH in the control run (dark blue) and for the RCP8.5 scenario BASE (red, light blue).

**References**

- Archer, S. D. et al.: Contrasting responses of DMS and DMSP to ocean acidification in Arctic waters, Biogeosciences, 10, 1893–1908, 2013.
- Park, K.-T., Lee, K., Shin, K., Yang, E. J., Hyun, B., Kim, J.-M., Noh, J. H., Kim, M., Kong, B., Choi, D. H., Choi, S.-J., Jang, P.-G., and Jeong, H. J.: Direct Linkage between Dimethyl Sulfide Production and Microzooplankton Grazing, Resulting from Prey Composition Change under High Partial Pressure of Carbon Dioxide Conditions, Environ. Sci. Technol., 48, 4750–4756, doi:10.1021/es403351h, 2014.
- Stefels, J., Steinke, M., Turner, S., Malin, G., and Belviso, S.: Environmental constraints on the production and removal of the climatically active gas dimethylsulphide (DMS) and implications for ecosystem modelling, Biogeochemistry, 83, 245–275, doi:10.1007/ s10533-007-9091-5, 2007.